# Simulation of a Custom-Made Temporomandibular Joint—An Academic View on an Industrial Workflow

**DOI:** 10.3390/bioengineering12050545

**Published:** 2025-05-20

**Authors:** Annchristin Andres, Kerstin Wickert, Elena Gneiting, Franziska Binmoeller, Stefan Diebels, Michael Roland

**Affiliations:** 1Applied Mechanics, Saarland University, Campus A4 2, 1. OG, 66123 Saarbrücken, Germany; annchristin.andres@uni-saarland.de (A.A.); kerstin.wickert@uni-saarland.de (K.W.); s.diebels@mx.uni-saarland.de (S.D.); 2KLS Martin SE & Co. KG, 78532 Tuttlingen, Germany; elena.gneiting@klsmartin.com (E.G.); franziska.binmoeller@klsmartin.com (F.B.)

**Keywords:** temporomandibular joint replacement, finite element analysis, biomechanics, custom-made prothesis, experimental prothesis testing, patient-specific model generation

## Abstract

Temporomandibular joint replacement is a critical intervention for severe temporomandibular joint disorders, enhancing pain levels, jaw function and overall quality of life. In this study, we compare two finite element method-based simulation workflows from both academic and industrial perspectives, focusing on a patient-specific case involving a custom-made temporomandibular joint prosthesis. Using computed tomography data and computer-aided design data, we generated different 3D models and performed mechanical testing, including wear and static compression tests. Our results indicate that the academic workflow, which is retrospective, purely image-based and applied post-operatively, produced peak stress values within 9–20% of those obtained from the industrial workflow. The industrial workflow is prospective, pre-operative, computer-aided design-based and guided by stringent regulatory standards and approval protocols. Observed differences between workflows were attributed primarily to distinct modelling assumptions, simplifications and constraints inherent in each method. To explicitly quantify these differences, multiple additional models were generated within the academic workflow using partial data from the industrial process, revealing specific sources of variation in stress distribution and implant performance. The findings underscore the potential of patient-specific simulations not only to refine temporomandibular joint prosthesis design and enhance patient outcomes, but also to highlight the interplay between academic research methodologies and industrial standards in the development of medical devices.

## 1. Introduction

Temporomandibular joint replacement (TMJR) represents a critical surgical intervention for patients with severe temporomandibular joint (TMJ) disorders, offering significant improvements in pain levels, jaw function and quality of life [1]. This procedure has become a standard of care for end-stage TMJ disease, particularly when conservative treatments fail to provide relief. The TMJ concepts system, along with other prosthetic systems, has shown promising results in restoring form and function, reducing pain and improving dietary consistency [2,3]. Studies have also found that total alloplastic TMJ prostheses are an efficient, safe and stable long-term solution, with significant improvements in mouth opening, pain relief and satisfaction with the surgery [4]. In addition, studies in humans and animals have shown that disturbances in normal masticatory function can lead to morphological and functional changes in the craniomaxillofacial system [5,6]. Despite all these advancements, the long-term implications of TMJR on masticatory function and overall biomechanics remain not fully understood [7,8]. Therefore, a clearer understanding of the biomechanical effects of TMJR is essential both to identify design shortcomings and to optimize the prostheses as well as the patient care.

The outcome of TMJ reconstruction is highly dependent on the kinematic performance, especially in the treatment of patients with internal derangement cases and in restoring form and function following the removal of failed TMJ implants [9]. In this context, understanding the biomechanics of the TMJR is important for improving the design and ensuring long-term success [10]. A kinematic study comparing the TMJ movements of healthy subjects with those of patients undergoing unilateral TMJ arthrotomy with metal fossa-eminence partial joint replacement showed significant differences in joint kinematics, underlining the need for biomechanical testing in design and development [11].

Finite element (FE) modelling has been widely used to evaluate the biomechanical behavior of TMJ prostheses [12,13]. Simulation-based studies have also shown that relevant shear forces can occur during mastication, which underpins the need for accurate musculoskeletal modelling in surgical planning [14]. It has also been shown that the development of customized temporomandibular fossa prostheses based on musculoskeletal simulations can significantly improve the outcome of total joint replacement procedures [15]. Simulation techniques, such as computed tomography (CT) scan data modelling, have been instrumental in the development and evaluation of custom prostheses for TMJ reconstruction [8]. These methods provide valuable insights into the performance and longevity of prosthetic joint replacements.

Joint simulators, like the VIVOTM joint simulator [16,17], provide a platform for evaluating the mechanics of joint replacements, including TMJ [18]. Computational modelling has also been used to study the fixation of fossa components in TMJR surgery, demonstrating the importance of personalized 3D-printed joint prostheses for optimal outcomes [19]. The biomechanics of the intact and implanted mandible during different biting tasks can reveal significant differences in joint movements and load distribution, which is helpful in identifying design limitations and improving possibilities.

In this study, we compare two different workflows for a selected real patient identified as a relevant clinical use case. The first workflow is the process of the manufacturer of the patient-specific TMJ prosthesis. This workflow is based on the pre-operative clinical image data, the surgical planning and the individualized design of the prosthesis together with the treating clinician. This procedure includes the mechanical tests and simulations required by the regulatory framework for the use of patient-specific TMJ prosthesis. This prospective workflow takes place entirely before the patient’s surgery.

In contrast, the academic workflow is based exclusively on post-operative clinical image data and takes place retrospectively to the patient’s surgery. In this context, the term “academic workflow” specifically refers to a simulation process relying solely on image segmentation and post-operative CT data, rather than proprietary pre-operative computer-aided design CAD data. As 3D-CAD data from manufacturers are not normally available in the academic environment or are only accessible with high barriers [20], academic simulation models are normally obtained exclusively from segmentation processes. To better assess the differences between the two concepts, one industry-driven and prospective and the other downstream and retrospective, we generated additional models based on the academic approach that link the segmented models with the manufacturer’s CAD data.

We show that the differences between the simulation results are small within the different assumptions and simplifications made. The study thus provides important insights regarding the possibilities of patient data-based simulations for TMJ prostheses, which hold substantial interest for the scientific community of this field. The main goal of this study was to show academic groups working purely retrospectively on image data, as the authors themselves normally do, a comparison with industry data-based simulations to enable them to better evaluate their own results.

The structure of the paper is organized as follows: Section 2 describes the patient data and methods used in both workflows to generate the computational models. Section 3 presents experimental results, including wear and static compression tests, as well as the simulation outcomes from both workflows. Section 4 discusses these results, focusing on the comparison between the industrial and academic approaches. Finally, Section 5 summarizes the key findings and conclusions drawn from the study.

## 2. Materials and Methods

### 2.1. Patient Case Data

A 35-year-old female patient presented with a fracture of the condylar process due to trauma. Initial treatment involved standard fracture management with mini plates. However, post-operative complications included an infection necessitating the removal of the condylar fragment. The patient exhibited reduced dentition, attributed to poor oral hygiene rather than trauma. Pre-operative planning was carried out using CT imaging, while post-operative assessments were conducted utilizing different imaging modalities, including CT, digital volume tomography (DVT) and orthopantomography (OPT). A custom-made TMJ prosthesis, consisting of a titanium alloy (Ti-6Al-4V) condyle and fossa component with ultra-high molecular weight polyethylene (UHMWPE) as joint surface, was inserted. Figure 1 provides an overview of the treatment and the personalized TMJ prosthesis.

The Ti-6Al-4V material used for the prosthesis offers a favorable surface for osseointegration, combined with high mechanical strength and ductility [21]. Due to the issues associated with a pure UHMWPE fossa, such as increased backside wear, poor surface for bone fixation, the increased chance for “cold flow” and a higher risk of osteolysis [19], we did not select a fossa made entirely of UHMWPE. Instead, UHMWPE was utilized as a sliding layer between two titanium alloy components, specifically as the joint surface on a Ti-6Al-4V base for the fossa component. The UHMWPE part is pressed onto a Ti-6Al-4V base body. This design combines the mechanical benefits of Ti-6Al-4V with the low-friction properties of UHMWPE at the articulating surface. Additionally, UHMWPE is recognized as a suitable material for articulating surfaces in joint implants according to DIN EN ISO 21534:2009. The condylar head’s design closely resembles the natural condyle, potentially leading to more natural movement for the patient. Custom-made prostheses outperform stock devices by providing a better fit, reducing operating time since no modifications are needed during surgery, and potentially decreasing wear and increasing implant longevity [19].

### 2.2. Industrial Workflow

The prosthesis manufacturer’s workflow described below is certified and officially approved by the authorities. After that, the workflow is frozen and has no room for changes, experiments or tests.

### 2.3. Segmentation and 3D Geometric Model Generation

The pre-operative CT scan was segmented using Mimics Medical 23.0 (Materialise, Löwen, Belgium). The CT data comprised 711 slices, each with a pixel size of 0.43 mm, a slice increment of 0.3 mm and a slice thickness of 0.5 mm. Masks were created for the mandible and teeth to generate stereolithography (STL) files for the finite element simulation. Additional masks for the facial nerve, cranium and soft tissue were generated to assist in the pre-operative planning process. These additional masks ensured safe distances from critical anatomical structures during the surgical procedure but were not included in the FE model used for biomechanical analysis. A standardized thresholding process, as suggested by Mimics, was applied, followed by visual control and manual post-processing as needed. The segmented data were then used to generate a 3D model, which was exported as an STL file. Case planning and the design of the TMJ prosthesis, including the placement of screws, were conducted using Geomagic Freeform Plus 2021.0.56 (3D Systems, Rock Hill, SC, USA). The mandible and teeth were merged into a single part, and the final design was exported as STL files for further processing and manufacturing.

### 2.4. Material Assignment and Meshing

The following material properties were used in the modelling: the part representing the mandible and the teeth were modelled as isotropic and linear elastic material [22] with a Young’s modulus of 10 GPa and a Poisson’s ratio of 0.3. The TMJ prosthesis and the screws are also modelled as isotropic linear elastic material with the values for Ti-6Al-4V given in [23] and in Table 1. The mesh generation resulted in a FE model with quadratic ten-node tetrahedral elements. The mesh information is given in Table 2.

### 2.5. Boundary Conditions

To ensure a high level of comparability between the two workflows, the boundary conditions used in the industrial workflow were also applied in the academic workflow. To apply the Dirichlet boundary conditions for the fixation of the models, two regions of interest (ROIs) were marked: one for the TMJ prosthesis and one for the opposite side of the mandible. These ROIs were exported as node sets during the meshing process. Another ROI was defined as tooth number 47, the first molar, to apply the bite force. In addition, ROIs were also marked to apply the boundary conditions given by the different muscles, cf. Figure 2.

Therefore, a standardized musculoskeletal simulation was performed using the Any-Body standard symmetric mandible model [24,25] within the AnyBody modelling system (AnyBody Technology A/S, Aalborg, Denmark). In this framework, muscle forces are computed using inverse dynamics combined with static optimization, which determines the net forces necessary to perform a given task. A biting task was simulated by applying a 450 N force at the first molar on the right side. This force was lower than the maximum bite forces reported by Ferrario et al. [26] and closely matched the loading conditions used by Pinheiro et al. [12], corresponding to 50% of the bite force values documented by Woodford et al. [27]. The resulting muscle forces (Figure 2) exceeded the values reported by Waltimo et al. [28], yet remained within the range documented in [12] and were accepted for use in the authorization process.

The simulation calculates both the magnitude and direction of each muscle force based on the geometry and kinematics of the underlying model.

Subsequently, the computed muscle forces were directly transferred to the patient-specific FE model. The force directions and magnitudes were used unaltered, and the muscle attachment points were mapped from the AnyBody model onto the patient-specific model. Muscles attached to the resected bone parts were removed to reflect the actual surgical conditions, ensuring the accuracy of the biomechanical environment post-resection.

To simulate the post-operative stabilization, the condyles were fixed. The boundary conditions are given in Figure 2. The illustration shows the positioning and the applied forces of the individual muscles. In addition, the fixed support of the condyles and the applied molar bite force are shown.

### 2.6. FEA Simulations

A standardized workflow was performed to generate the simulation model. This industry process ensured consistency and reproducibility across simulations and cases. To accurately simulate the biomechanical interactions, contact was established using the following constraints: (1) screws and prosthesis: bonded with a multi-point constraint and (2) screws and mandible: bonded with a multi-point constraint. The modelled bonded contact for the screw interface replicates the fixed connection between screw and bone material for an ingrown condition as well as between screw and plate for a locked thread connection. This condition is known to effectively replicate the global deformation behavior and the load transfer through the fixation construct when local stresses in the threads and the interface are not considered [29,30]. These constraints accurately reflected the clinical conditions and ensured the reliability of the simulation results. The contact between prosthesis and mandible was considered as part of the approval process of the software in the verification step, but as there was no penetration of the bone, it was decided not to consider this contact in the frozen workflow.

### 2.7. Mechanical Testing

The TMJ prosthesis by KLS Martin (KLS Martin SE & Co. KG, Tuttlingen, Germany) was realized as a custom-made device made of Ti-6Al-4V and UHMWPE. The average wear of the prosthesis was tested on the robotic system HORST600 (fruitcare robotics GmbH, Konstanz, Germany). The wear test was conducted with reference to the standard ISO 14243-1. The prosthesis components were moved against each other in a cyclic movement consisting of 5 Mio cycles with a constant load representing a normal chewing force of 20 N [31,32]. This movement consisted of a combination of a 12° rotation and a translational movement of 8.5 mm anterior/posterior and 0.7 mm medial/lateral of the condyle. The wear was measured gravimetrically, and the net mass loss and average wear rate were calculated in accordance with ISO 14243-2.

For the static compression tests, the worst-case design was tested on the standard testing device Z020 (ZwickRoell GmbH & Co. KG, Ulm, Germany) with five testing samples. The native bone was replaced with a patient-specific mandible model made of polyamide (PA), produced via additive manufacturing on EOS Formiga P110 (EOS GmbH, Krailling, Germany). The TMJ prosthesis, including the titanium condyle component, was fixed to this PA mandible model in a closed jaw position (0° opening) and loaded until failure. PA was chosen as the bone substitute material because it allowed for accurate replication of the patient’s mandible geometry and consistent fixation of the implant [33,34]. We acknowledge that the use of PA may affect contact stiffness compared to natural bone or titanium. Nonetheless, it provided a practical and reproducible means to evaluate the prosthesis under standardized testing conditions [35,36]. The load was applied 30 percent on the contralateral and 70 percent on the ipsilateral side of the defect, as has been done in similar test setups [37]. After testing, the samples were scanned with the 3D scanner ATOS Core (Carl Zeiss Industrielle Messtechnik GmbH, Oberkochen, Germany) to compare the state of the samples before and after testing to check for deformation.

### 2.8. Academic Workflow

#### Segmentation and 3D Geometric Model Generation

The individual digital imaging and communications in medicine (DICOM) image stack of the patient’s post-operative CT scan was used to create a geometric model. The tomogram consists of 434 images with a constant voxel, i.e., volumetric pixel, size of 0.3 mm in all three spatial directions. The image stack was segmented into different masks: (1) the mandible, (2) TMJ prosthesis, (3)/(4) the two mandibular central incisors, numbers 41 and 31 in the FDI world dental federation notation (ISO 3950), (5)/(6) the two mandibular lateral incisors, numbers 42 and 32, (7)/(8) the two mandibular canines, numbers 43 and 33, (9) the lower left mandibular first premolar, number 34, (10) the lower right mandibular second molar, number 47 and (11) a past treatment of tooth number 43, i.e., mask (7). The segmentation process was realized in the software ScanIP (ScanIP V24-06 - Academic, Synopsys, Mountain View, CA, USA) and started with an adaptive thresholding procedure for every mask. After that, for each mask, the morphological filters “island removal”, “cavity fill” and “fill gaps” with a priority order were applied resulting in a first segmentation. Then for each mask, the results were visually controlled and manually post-processed to provide a high-quality segmentation without detectable problems, cf. Figure 1A–D for the segmentation result.

Since the industrial workflow uses computer-aided design (CAD) data of the treatment rather than a segmented model of the TMJ prosthesis, we decided to create four models within the academic workflow to gradually transition from a purely segmented model toward the industrial workflow and to highlight differences in the results. These four models are:

Model No. 1: A standard version based solely on the segmented data (bone, teeth and prosthesis).

Model No. 2: A segmented bone-teeth model in which the prosthesis was replaced by the CAD-data from the prosthesis design representing the actual treatment.

Model No. 3: A segmented bone-teeth model in which the prosthesis was replaced by the CAD-based prosthesis design (the same as Model No. 2) but including two additional “dummy screws” (described below) from the surgical pre-operative planning, i.e., the model used in the industrial workflow.

Model No. 4: A further adaptation (described under subsection FEA simulations) that incorporates contact boundary conditions and a more simplified model, closely reflecting the industrial approach.

Here, the term “dummy screw” refers to two screws that were planned to be used during the pre-operative consultation with the clinician but could not be placed intraoperatively due to surgical constraints. Since the industrial workflow (based on pre-operative data) includes these screws, whereas the academic workflow (based on post-operative data) generally does not, both variants were simulated for consistency and comparability. This approach enables a thorough comparison of all models via biomechanical FE analysis.

For Models No. 2, No. 3 and No. 4, the CAD data of the two prosthesis designs were imported into ScanIP and manually aligned with the segmentation using the “position and orientation” tool. Subsequently, the surface deviation between the CAD-based prosthesis and the segmented prosthesis was computed via ScanIP’s surface deviation function, which uses sampling points and the closest-point method. This procedure facilitates assessment of the segmentation quality.

### 2.9. Material Assignment and Meshing

The assignment of material properties proceeded as follows. The mandible was modelled as an isotropic, non-homogeneous, linear elastic material. In this context, “non-homogeneous” indicates that the Young’s modulus was computed for each mesh cell based on the corresponding Hounsfield units (HU) from the CT data, with each mesh cell treated as homogeneous at that local scale. All other materials were defined as isotropic, homogeneous and linear elastic. The complete set of material parameters and HU-based mappings can be found in Table 1.

Because no calibration phantom was used, the CT scans were calibrated via linear interpolation between two known data points, as described by Zannoni et al. [38]. Following the approach of Gray et al. [39], the mean radiographic densities of water and air served as reference points. Subsequently, the calibrated CT data were mapped linearly to CT density in accordance with the values presented in Table 1. The Young’s modulus for each mesh cell was then calculated by applying the power–law relationship also provided in Table 1.

For the masks representing the teeth, we used the mean values reported in Pinheiro et al. [12] for dentin and enamel, referencing Murphy et al. [40] and Zhou et al. [41]. The distinction between dentin and enamel was established according to the respective HU in the CT data. For the treatment area of tooth number 43, we assigned the material parameters of enamel, since this region does not influence the FEA outcome and is treated equivalently to healthy tooth structure. The TMJ prostheses were assigned material properties from Pinheiro et al. [12] and Korioth et al. [42]. All material assignments were performed in ScanIP, where the Young’s modulus and Poisson’s ratio were stored for each mesh cell in the computational grid.

Regarding the meshing strategy, we selected quadratic, ten-node tetrahedral finite elements (C3D10) with straight edges for all masks. Volume meshing was carried out in ScanIP using a coarseness factor of “−10” for all masks. The resultant mesh details are summarized in Table 2. Following the meshing step, all data were exported as input files for the Abaqus CAE/2022 simulation environment (Dassault Systèmes, Vélizy-Villacoublay, France).

The four finite element (FE) models developed and employed in the academic workflow are illustrated in Figure 3. Model 1 was generated solely from segmentation data and served as baseline geometry. Model 4 was specifically developed for contact simulations to better replicate prosthesis–screw–bone interactions. Models 2 and 3 integrated segmented mandibular and dental anatomy with the CAD-based prosthesis: Model 2 corresponds to the finalized clinical treatment scenario, while Model 3 reflects the pre-treatment planning stage, including two additional dummy screws for simulation purposes.

### 2.10. FEA Simulations

To biomechanically simulate the patient’s current treatment, we adopted a simulation workflow previously employed for patient-specific analyses and virtual testing of bone-implant systems [43,44,45]. In the present work, we partially transferred this established workflow to the patient-specific evaluation of a TMJ prosthesis.

All simulations were performed in Abaqus on a standard workstation (Intel^®^ Core™ i9-9920X CPU @ 3.50 GHz, 128 GB RAM, 64-bit Windows 10 Pro). The simulations were run sequentially in a queue managed by a custom Python script (Abaqus Python 3.7). Result files in the Abaqus output database (odb) format were post-processed with Python-based in-house software. Consistent with the previously cited workflows, contact interactions were initially omitted. This approach is typical in academic workflows, which often rely on segmented datasets that produce large, highly refined meshes. Such fine meshes substantially increase computational time, particularly when the prosthesis is also generated by segmentation. This approach can introduce geometric inaccuracies compared to a CAD-based representation.

Nevertheless, recognizing the importance of contact conditions on the biomechanical performance of TMJ prostheses, we conducted two additional simulations mirroring the industrial workflow: one with contact and one without. To reduce computational expense, we created a simplified mesh by merging the mandible and all teeth into a single mask and assigning a uniform elastic modulus of 10 GPa. This simplified model, introduced above as Model No. 4, was derived from the CAD representation of the prosthesis, including the two dummy screws.

In the contact simulation, “surface-to-surface contact” was specified in Abaqus, with a friction coefficient of 0.3 [46] for tangential behavior and “hard contact” for normal behavior. The contact surfaces were automatically generated in ScanIP and exported via the Abaqus input file. Figure 4 depicts the results from these two simulations, illustrating the potential influence of contact interactions on the overall biomechanical analysis.

## 3. Results

### 3.1. Mechanical Testing

Figure 5 shows the average wear rate recorded during testing. The graph shows a peak for both components after the first 500,000 cycles, followed by a decrease. These results align with general findings that indicate a “running-in” phase, followed by a “steady-state” phase [47]. Calculating the volume loss based on the number of cycles results in a wear of 0.624 mm^3^/10^6^ cycles for the fossa components and 0.0688 mm^3^/10^6^ cycles for the condyle component. The results for the fossa component are similar to the volume losses found in comparable devices, such as the fossa component of the Groningen prosthesis (0.65 mm^3^/year) [48,49]. However, the results for the condyle component are less comparable due to a lack of data on the wear of metal components in total joint replacements. Therefore, the results are compared to metal-on-metal pairings in hip prostheses, which showed wear between 0.24 mm^3^/10^6^ cycles and 1.84 mm^3^/10^6^ cycles [50]. In both cases, the custom-made TMJ prosthesis demonstrated wear rates below those of other total joint replacements.

In the static compression test, no differences were observed between scans before and after testing, indicating no deformation in the implants, with only minimal deformation observed in the fossa. The maximum loads reached 1540 N on average. Failure occurred in the tested samples due to screws being pulled out of the polyamide jaw at the maximum loading of the test setup, not due to damage to the implant itself. Figure 6A illustrates the typical load case, Figure 6B,C show a before/after comparison for the worst case of the Fossa component (B) and the Condyle component (C). For both components, the deviation of the surface area stays below 0.3 mm. The Fossa component (B) has a slight indentation at the point where the condyle was pressed into it. The condyle component does not show any deformation from the compression test. Overall, the results demonstrate the robustness and wear resistance of the custom-made TMJ prosthesis compared to other devices.

### 3.2. Segmentation and 3D Geometric Model Generation

Figure 1 presents the segmentation results from both workflows. Figure 1A shows a threshold-based representation of the 3D tomogram to provide an overview of the patient situation after surgery. Figure 1B–D display the results of the academic workflow, whereas Figure 1E,F show the industrial workflow. In Figure 1B, a lateral view of the segmentation is depicted, while Figure 1C,D show an anterior view of the 3D model. Figure 1D is partially transparent to highlight the segmented teeth and screws. Figure 1E depicts the segmentation result from the industrial workflow alongside the generated CAD data of the planned treatment. Finally, Figure 1F provides an overview of this planned treatment.

A direct comparison of the segmented mandible bones was omitted because different software systems and masks were used. For a valid comparison, the segmented masks in the academic workflow would have to be converted into STL surfaces to align with the STL data from the industrial segmentation. However, especially for small structures such as teeth, artifacts and minor errors occur during file conversion, affecting the results. Consequently, no comparison was conducted for the segmented bones. Instead, as described below, the focus was on comparing the TMJ prosthesis.

Figure 7 displays the segmentation results for the TMJ prosthesis. Figure 7A,B show the prosthesis based on the CAD data from the surgical planning provided by the industrial partner. Figure 7C illustrates the corresponding segmentation outcome in the academic workflow, which captures the prosthesis geometry reasonably well, despite minor artifacts. Figure 7D shows the same model after applying recursive Gaussian smoothing with an isotropic value of 1.5 pixels. Although smoothing reduces artifacts, it introduces new challenges for the segmented screws. Because the screws in this application are relatively small (in this study, the voxel size is 0.3 mm^3^, and thus each screw consists of only a few voxels), even adding or removing one or two voxels becomes visibly significant in the 3D model. Consequently, recursive Gaussian smoothing was not applied to preserve the screws. Figure 7E shows the screws on the back side of Figure 7C. This configuration was used for Model No. 1 in the academic workflow, with each screw visually inspected and manually refined when needed. Another difference between the CAD-based surgical planning and the segmented TMJ prosthesis is as explained above the presence of two so-called “dummy screws” in the planning (Figure 7B) and their absence in Figure 6E.

Figure 8 presents the distance map (in millimeters) between the CAD data of the TMJ prosthesis and its segmented counterpart, visualized on the CAD model. The distances are mostly below 1 mm, with the greatest deviations around the two dummy screws, which are only present in the CAD data but not in the current treatment. These small discrepancies indicate that the segmentation results are largely accurate, consistent with the similarity observed in the subsequent simulation outcomes.

### 3.3. Biomechanical FE Simulations

The industrial partners in this study normalize their simulation results against an in-house material limit, specified in MPa and calculated as von Mises stress. This material limit incorporates requirements for the selected material, the manufacturing process and a safety factor. Because these details are proprietary, all academic workflow results are normalized relative to the industrial workflow’s maximum stress value, without disclosing the latter.

Figure 9 shows the academic workflow’s simulation results. Figure 9A illustrates the outcome for the fully segmented model, exhibiting a normalized maximum von Mises stress of approximately 1.20, about 20 percent above the industrial workflow reference. This maximum is also roughly 10 percent higher than those seen in the two other academic simulations, highlighting differences between a segmented representation of the treatment and its CAD-based version.

Figure 9B,C both show maxima of around 1.11 and 1.09, respectively. The slight variation mainly arises from differences in mesh generation and the manual alignment of the prosthesis. These results also indicate that the two dummy screws present only in the planning phase have no relevant impact on the performance of this custom-made TMJ prosthesis. All three academic models display their maximum stress at the same prosthesis location for the analysed molar bite force.

Figure 4 presents the industrial workflow’s simulation results. Under the specified boundary conditions, about one-fifth of the material limit is reached. Notably, the maximum stress region differs from that observed in the academic workflow simulations, partly due to how boundary conditions are applied. In the industrial workflow, they are derived from a corresponding model, whereas in the academic workflow, muscle attachment sites must be manually defined in ScanIP or Abaqus. Additionally, minor differences in the coordinate systems of the various software platforms may contribute to these deviations.

Figure 10 compares simulations with and without contact conditions between the prosthesis and the mandible for the simplified version Model No. 4. When contact conditions are excluded (Figure 10A), slightly higher peak stresses are observed. Including contact (Figure 10B) reduces the peak stress by nearly 9 percent. Although this difference is small in the current application, it may be significant in other scenarios. Decisions regarding whether to include contact conditions should thus be made on a case-by-case basis.

## 4. Discussion

This study demonstrates that two simulation workflows, each relying on different preconditions and assumptions, can nevertheless yield comparable results. The observed differences can be explained by the different assumptions made in each workflow, highlighting opportunities for future investigations.

A key factor affecting the academic workflow’s outcomes, as noted by Pinheiro et al. [12], is the absence of a dedicated material mapping law for the mandible in literature. We therefore adopted the material mapping introduced by Pinheiro and colleagues. Although their choice is well-reasoned, addressing this limitation would require extensive experimental work, which lies beyond the scope of the current study. This issue is closely linked to the industrial workflow’s use of a single set of material parameters; indeed, differences in the level of detail for bone stiffness likely contribute to the observed variations.

Another closely related aspect is the segmentation process itself, which is subject to error. As Irshad et al. [51] showed, repeated segmentation may yield inconsistent results, and outcomes can also depend on the software employed. Moreover, boundary conditions could potentially be enhanced by using alternative meshing techniques, such as iso-topological meshes [52], which were not part of either workflow here but may offer advantages in future studies.

Further consideration is the handling of contact between the prosthesis and screws, and between the prosthesis and bone. In the academic workflow, no contact was defined, as such a distinction is not feasible when both prosthesis and screws are purely segmented. By contrast, in the industrial workflow, contact conditions are included, leading to a more realistic stress distribution.

The manual alignment of the 3D prostheses also affects the results. Minor deviations arise both within the academic workflow (across different models) and between the academic and industrial workflows. In addition, small discrepancies exist between the planned prosthesis positioning and its actual intraoperative placement, particularly for the screws and their insertion angles. These deviations from the ideal 3D CAD model are unavoidable in surgery but were found to have only a minimal impact on the simulation outcome, as illustrated by the simulations in the academic workflow. They also confirm that excluding the two unused “dummy screws” has little influence on the peak stress.

In addition to these technical considerations, our findings have important implications for patient care and the future design of TMJ prostheses. First, the simulation-based approach presented here can enhance clinical outcomes by enabling more personalized treatment strategies. By providing a detailed, patient-specific biomechanical analysis, such methods have the potential to optimize implant design for individual anatomical variations, potentially reducing surgical time and minimizing post-operative complications.

Moreover, the retrospective simulation method offers a valuable tool for future design improvements. It allows researchers and clinicians to evaluate the performance of prosthesis designs using post-operative data, thereby identifying design limitations and areas for enhancement that may not be apparent during pre-operative planning. This retrospective analysis also enables academic groups to develop and test alternative designs independently, without relying on proprietary CAD data from manufacturers. While this independence can spur innovation by fostering novel design approaches, it also carries certain risks. For instance, designs developed solely through academic simulation may not fully consider manufacturing constraints or regulatory requirements and therefore must be rigorously validated before clinical adoption.

While our study is based on a single clinical case, we view this work as a proof-of-concept that establishes the feasibility of comparing simulation workflows in TMJ replacement. The patient included in this report was specifically selected because she underwent a unilateral TMJ replacement with comprehensive pre- and post-operative imaging data. Moreover, she was one of the first patients to receive a customized TMJ prosthesis, limiting our ability to increase the number of cases at this stage. The limited number of cases restricts the generalizability of our findings, and we acknowledge that further validation with a larger cohort is required to statistically substantiate these preliminary results. In the future, we plan to conduct a simulation-based study involving a larger patient cohort, allowing for statistical analysis and improved generalizability. Nevertheless, the insights gained from this single-case comparison offer valuable guidance for refining simulation models and identifying key design parameters that could influence prosthesis performance.

## 5. Conclusions

This study demonstrates that academic and industrial simulation workflows for TMJ prostheses, despite differing in data sources, assumptions and timing, can yield comparable biomechanical outcomes. The academic workflow, relying solely on retrospective, image-based modelling, was able to reproduce key features of the industrial, pre-operative CAD-driven process. Differences observed between workflows were primarily due to variations in material assumptions, segmentation detail and modelling of contact conditions.

In terms of clinical significance, our findings suggest that simulation-based approaches have the potential to enhance patient care by enabling personalized treatment strategies for TMJ replacement. A detailed, patient-specific biomechanical analysis can help optimize implant design to match individual anatomical and functional requirements, potentially reducing surgery time and minimizing post-operative complications. Moreover, the retrospective simulation method offers a means to refine prosthesis designs based on real-world clinical outcomes, complementing the manufacturer-based pre-operative planning process. However, it is important to recognize that while academic-driven simulation models provide an independent avenue for innovation, they also carry risks if designs are developed without fully integrating manufacturing constraints or regulatory requirements. Therefore, rigorous validation and collaboration with industry partners remain essential for the successful clinical translation of these findings.

Overall, our results underscore the potential of simulation-based methodologies to drive future improvements in TMJ prosthesis design and ultimately enhance clinical outcomes. Future studies with a larger patient cohort will be critical to further explore these parameters and validate the clinical applicability of our simulation approaches. Nevertheless, our results suggest that simulation-based approaches have the potential to enhance clinical outcomes by enabling more personalized treatment strategies and improving the design of custom-made TMJ prostheses.

## Figures and Tables

**Figure 1 bioengineering-12-00545-f001:**
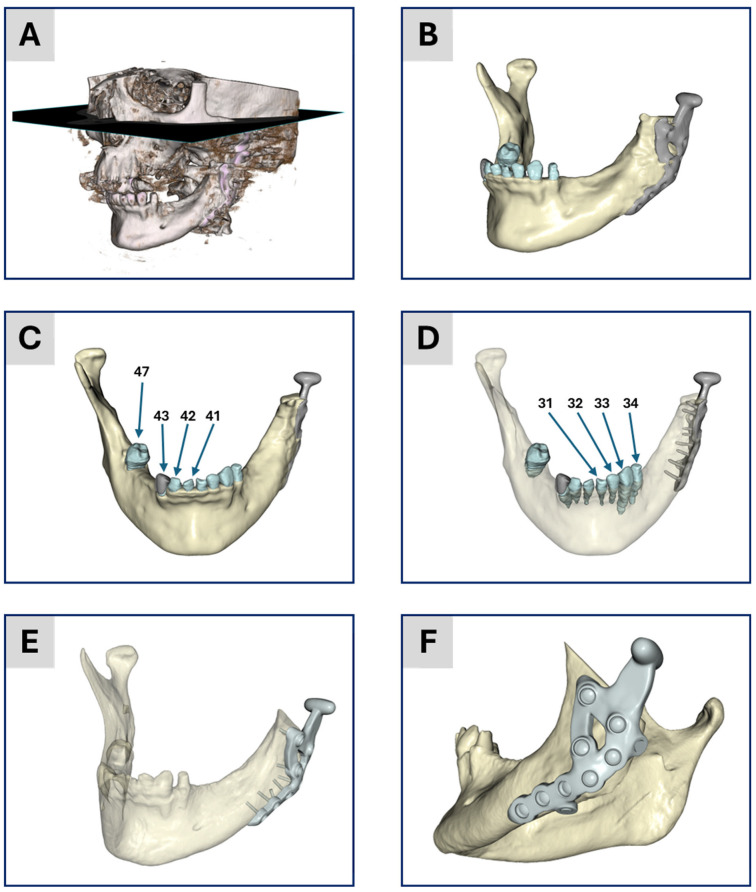
Illustration of the segmentation and model generation process. (**A**) Threshold-based image of the 3D tomogram giving an overview of the patient situation. (**B**–**D**) The results from the academic workflow and (**E**,**F**) the results from the industrial workflow. (**B**) Lateral view of the segmentation result. (**C**) Anterior view of the geometric 3D model after segmentation. (**D**) Same illustration as (**C**) with transparent bone to show the segmentation of the teeth and the position of the screws. The numbers correspond to the numbering of teeth according to the FDI world dental federation notation. (**E**) Segmentation results together with the CAD data of the treatment from the surgical planning. (**F**) Overview of the CAD data for visualization of the treatment. Items (**A**–**D**) were generated based on data from ScanIP, and items (**E**,**F**) were generated using data from Mimics; see the Segmentation and 3D Geometric Model Generation section of the two workflows below for further details.

**Figure 2 bioengineering-12-00545-f002:**
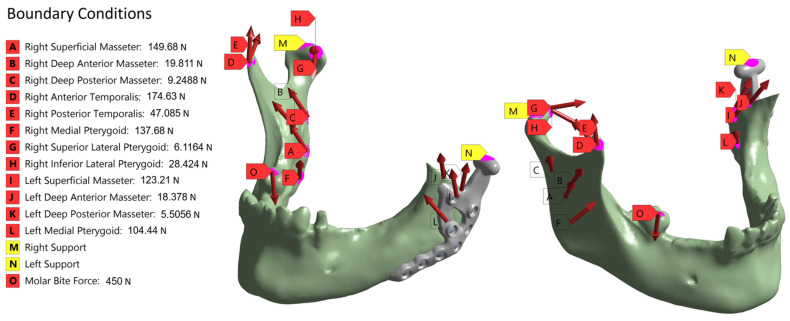
Illustration of the set-up of the boundary conditions based on the musculoskeletal simulation. In addition, the two supports for the fixed Dirichlet boundary conditions are shown in purple as well as the area for applying the molar bite force. The left-hand column contains all the values that were applied for the individual muscle forces.

**Figure 3 bioengineering-12-00545-f003:**
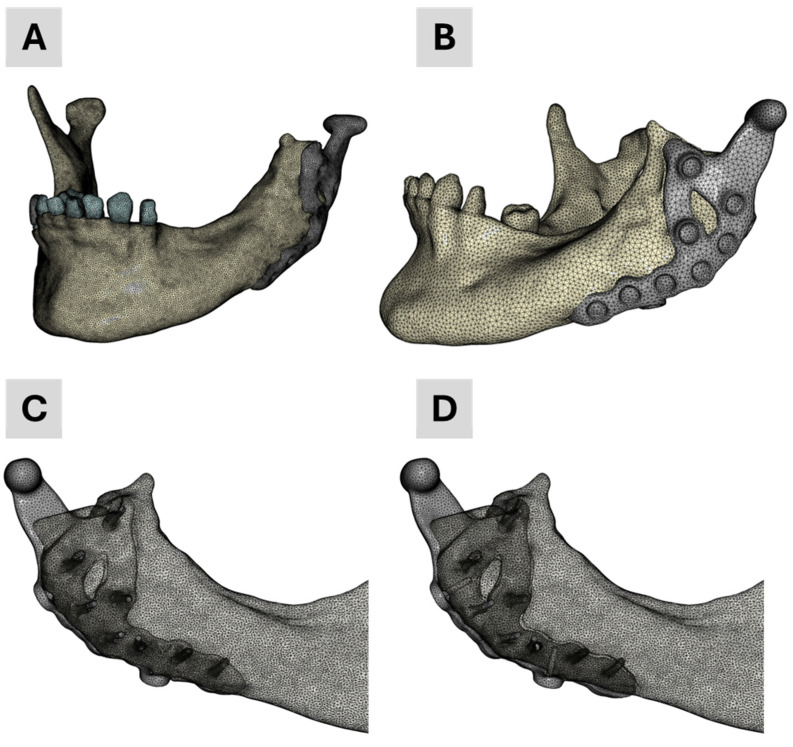
Finite element meshes of the four models used in the academic workflow: (**A**) Model 1—segmentation only; (**B**) Model 4—used for contact simulations; (**C**) Model 2—final treatment model combining segmentation and CAD prosthesis; (**D**) Model 3—planning stage model with two additional dummy screws.

**Figure 4 bioengineering-12-00545-f004:**
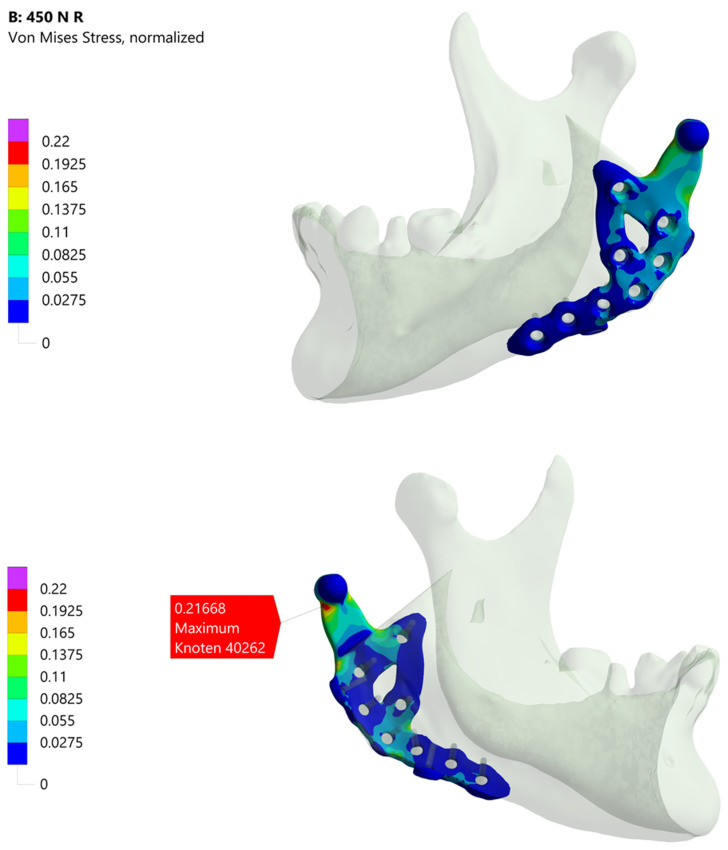
Illustration of the simulation result of the industrial workflow. The images show the normalized von Mises stress distribution of the prosthesis for the given molar bite force.

**Figure 5 bioengineering-12-00545-f005:**
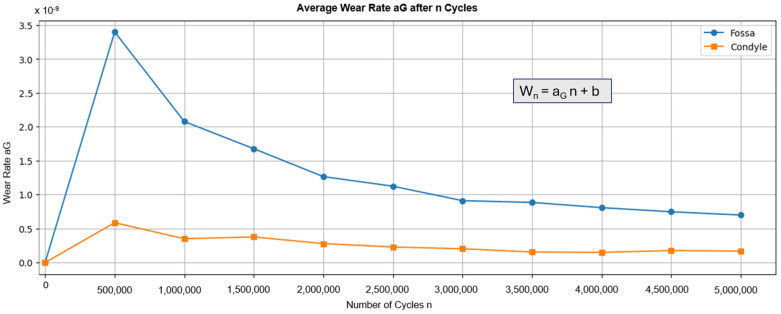
Graph displaying the average wear rate of the fossa and condyle components during wear testing in steps of 500,000 cycles. The wear rate was calculated according to ISO 14243-2, chapter 4.6.5. Here, W_n_ is the net loss in mass after n cycles and b is a constant.

**Figure 6 bioengineering-12-00545-f006:**
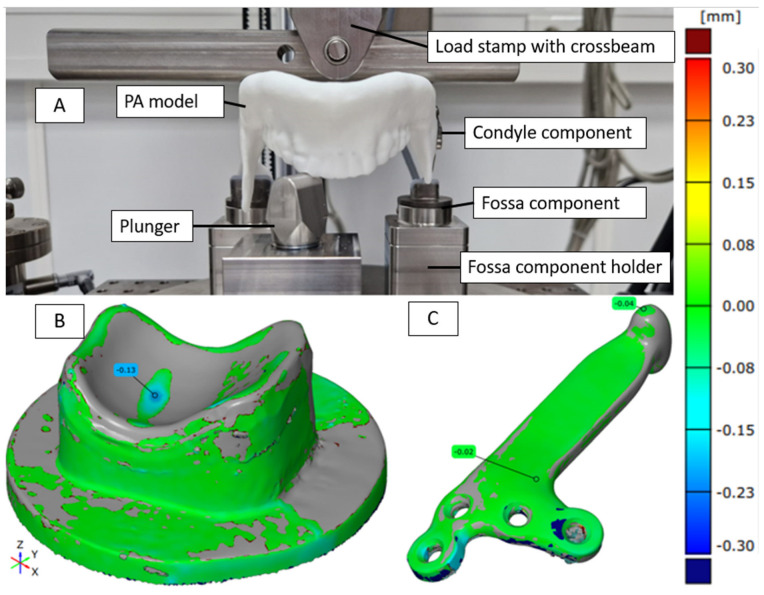
Illustration of the static compression test of the TMJ prosthesis by KLS Martin. (**A**) The test setup used for the compression test. (**B**,**C**) An example of a comparison of the surface of a fossa (**B**) and condyle (**C**) component before and after loading.

**Figure 7 bioengineering-12-00545-f007:**
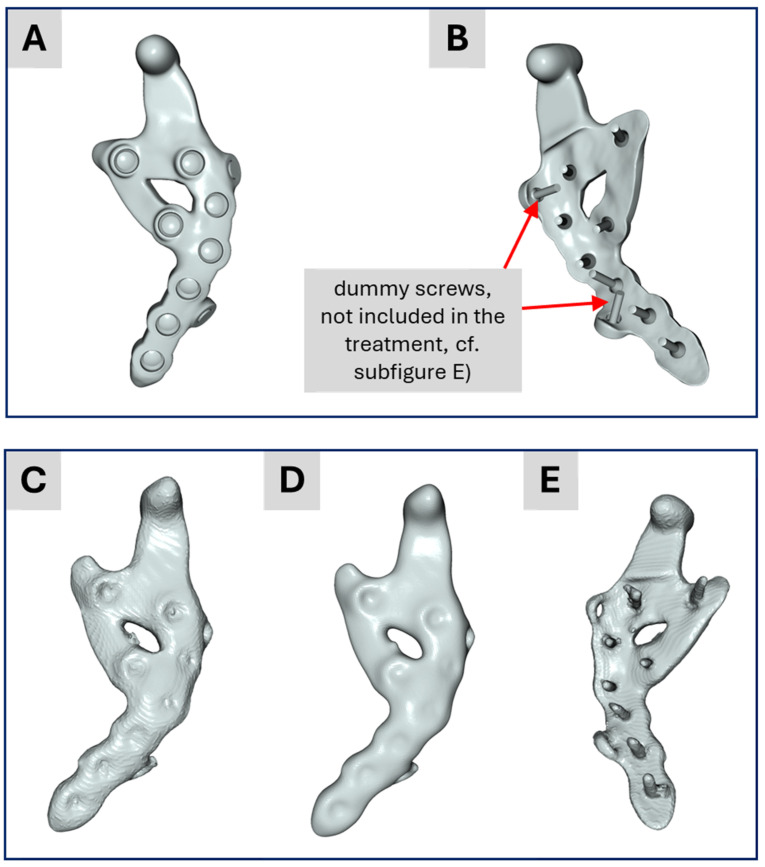
(**A**,**B**) Illustration of the CAD data representing the custom-made implant. (**C**) Result of the academic segmentation process for the implant. (**D**) Implant from (**C**) after applying a recursive Gaussian smoothing. (**E**) Backside of the segmented implant from (**C**).

**Figure 8 bioengineering-12-00545-f008:**
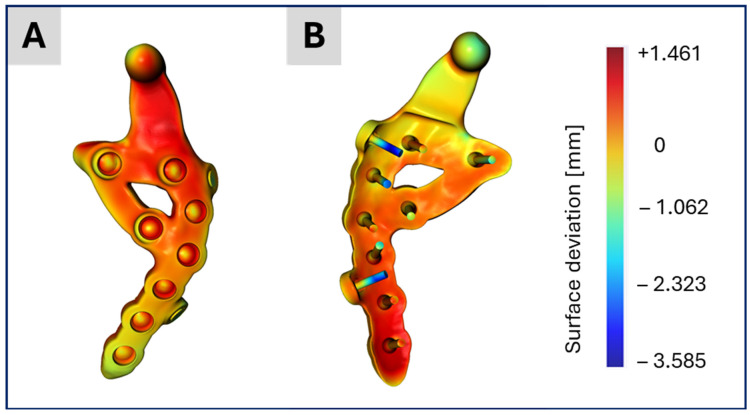
Illustration of the surface deviation in millimeter of the CAD-based TMJ prosthesis after their placement on the segmented TMJ prosthesis in the CT image stack. The highest differences are at the two dummy screws which are of course missing in the segmented TMJ prosthesis. (**A**) shows the lateral view on the prosthesis and (**B**) the backside and the screws.

**Figure 9 bioengineering-12-00545-f009:**
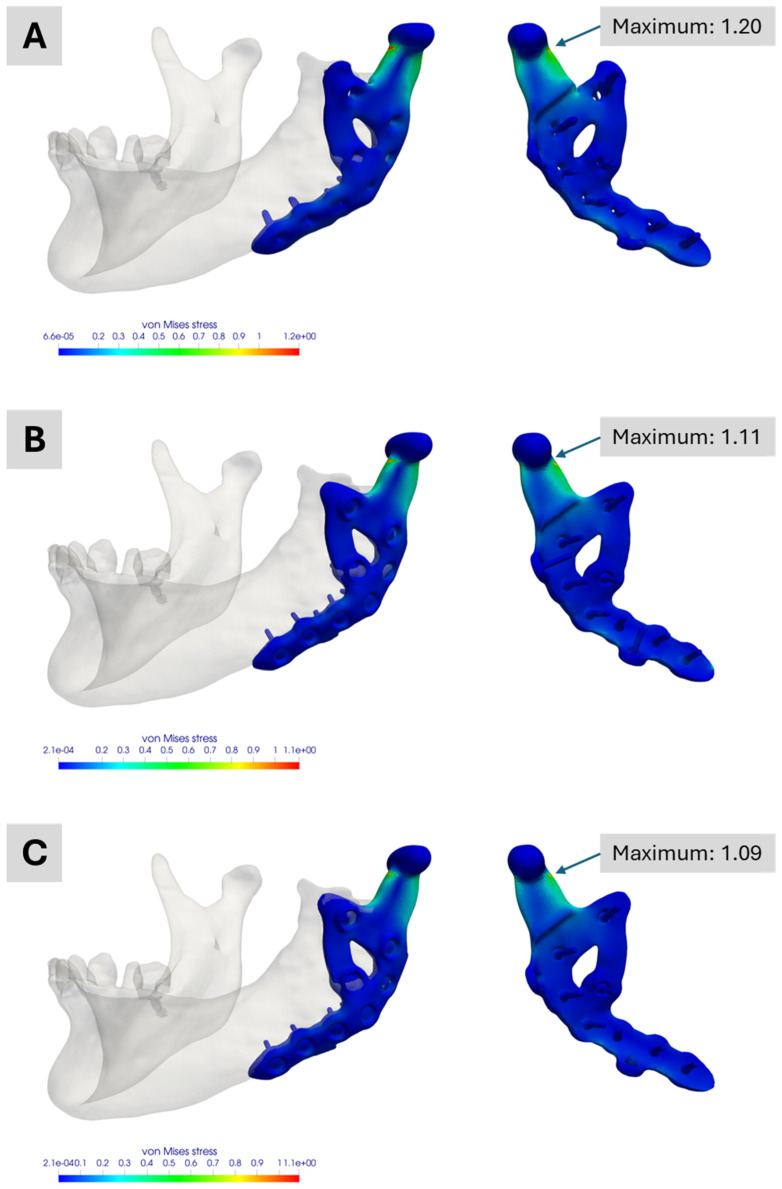
Simulation results of the Models No. 1, No. 2 and No. 3 in the academic workflow. (**A**) Model No. 1 based on the segmentation results. (**B**) Model No. 2 combining segmentation and CAD data of the prosthesis for real treatment. (**C**) Model No. 3 combining segmentation and CAD data of the prosthesis with the two additional screws. All images show the von Mises stress distribution of the prosthesis for the considered molar bite force.

**Figure 10 bioengineering-12-00545-f010:**
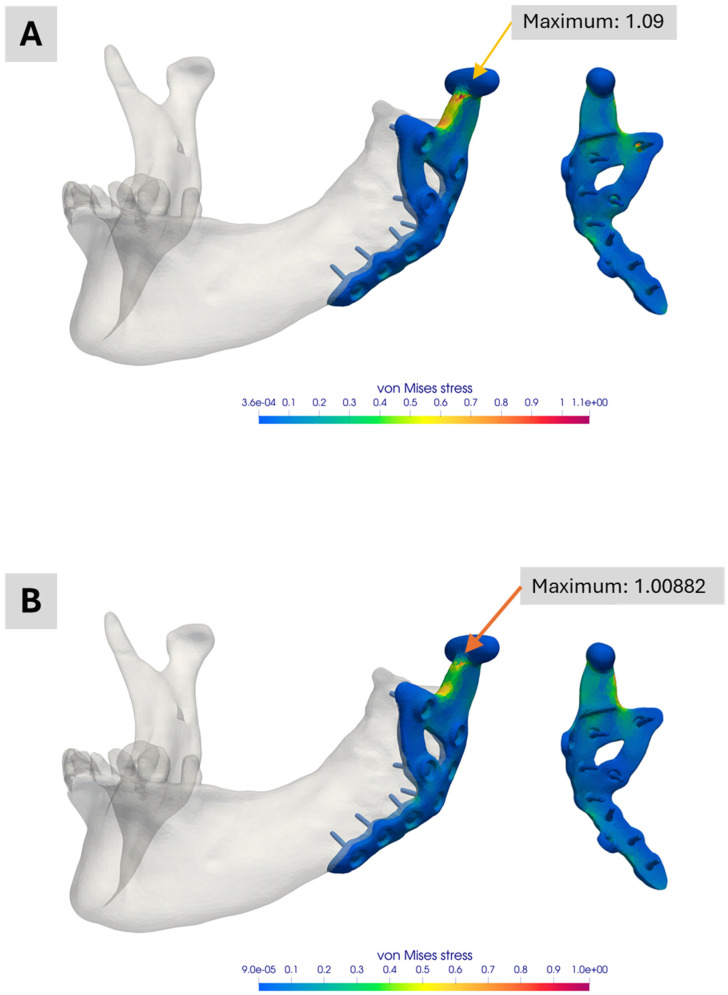
Illustration of the simulation result comparing the influence of contact conditions between prosthesis and mandibular. (**A**) Normalized von Mises stress distribution for the simplified version of Model 3 without contact conditions. (**B**) Normalized von Mises stress distribution for the same model with contact conditions. In the case of contact conditions, the peak stress is almost 9 percent lower.

**Table 1 bioengineering-12-00545-t001:** This table is essentially taken from Pinheiro et al. [12] and has only been adapted here to our application and its underlying data.

Mask	Young’s Modulus E (MPa)	Poisson’s Ratio	CT Values (HU)	CT Density ρ (kg/m^3^)	Young’s Moduli Law
Mandibular, Trabecular Bone	180–380	0.300	−1000–500	0 ≤ ρ < 1000	E = 0.0004 ρ^2.01^
Mandibular, Cortical Bone	11,300–22,900	0.300	501–1500	1001 ≤ ρ < 2000	E = 0.005 ρ^2.01^
Dentin	24,535	0.300	≤2000	-	-
Enamel	39,605	0.300	>2000	-	-
Ti–6Al–4V	113,800	0.342	-	-	-
Bone	10,000	0.3	-	-	-

**Table 2 bioengineering-12-00545-t002:** Number of nodes and mesh cells of all models from both workflows.

		Number of Nodes	Number of Mesh Cells										
					Teeth								
Workflow	Model		Mandibular	TMJ Prosthesis	No. 31	No. 32	No. 33	No. 34	No. 41	No. 42	No. 43	No. 47	Treatment of No. 43
Academic	Model No. 1	1,055,863	535,018	75,635	7295	6147	10,902	7024	5641	7763	5391	17,721	3485
	Model No. 2	1,069,302	544,807	76,398	6679	6152	10,908	6962	5886	7616	4986	17,377	3351
	Model No. 3	1,056,745	536,552	74,826	6701	5956	10,941	7192	5710	7742	5347	17,688	3560
	Model No. 4	157,867	72,602	19,148	-	-	-	-	-	-	-	-	-
Industrial	-	975,770	164,306	435,695	-	-	-	-	-	-	-	-	-

## Data Availability

The original contributions presented in the study are included in the article, further inquiries can be directed to the corresponding author. Researchers who wish to request access to data should send an email, with a clear indication of the research purpose. Every request must be reviewed by the responsible institutional review boards, considering the risk of patient reidentification and the compliance with the applicable data protection rules.

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
