# Peer review of "Simulation of a Custom-Made Temporomandibular Joint—An Academic View on an Industrial Workflow"

_bioengineering, 2025, doi:10.3390/bioengineering12050545_

Round 1
Reviewer 1 Report
Comments and Suggestions for Authors
This reviewer is an Oral and Maxillofacial Surgeon and not an engineer, so I am not able to comment in detail about the engineering aspects of this study, although I do understand the processes which have been used. This study uses prosthesis data from one patient who underwent a unilateral TMJ replacement.
My comments are centered around the Discussion. I would like to see comment about how this work could influence patient care in the future ? How exactly might this retrospective method influence future design ? Is it mainly a way for academics to develop alternative designs without direct assistance from a manufacturer who may withold commercially sensitive data ? Are there any risks in doing that ?
I note that the design used in this study differs significantly from that of the TMJ Concepts system and from the Biomet-Zimmer system. It resembles an Australian device referred to in reference 19. The authors may not know that there is a series of ongoing legal cases in Australia relating to the surgeon who helped to develop that device, which itself is no longer in current use. My advice would be to remove reference 19.
Finally please can the authors check the journal instructions regarding style of references. Most have doi data rather than the final publication data.
Author Response
We thank the reviewer and have answered everything in the attached pdf file.

Reviewer 2 Report
Comments and Suggestions for Authors
- I am unable to comprehend what the authors mean by “an academic workflow”? Is it the 3D simulation that they are referring to about? If so, the term should be suitably edited.
- What was the basis for patient selection in this report? This has to be clearly explained.
- Authors must justify why comparison between the different workflow models were based on just one case. This makes us question the validity of the evaluated data.
- The discussion could elaborate a little bit more about the clinical significance of the study and how the present study results should be extrapolated in clinical cases of TMJ replacement.
Author Response

(The authors gave the same response as above.)

Reviewer 3 Report
Comments and Suggestions for Authors
1. The Young’s modulus of the mandible in the industrial modeling is incorrectly referenced. The cited source does not explicitly state that the mandibular Young’s modulus should be set to 10 GPa. Instead, the material model mentioned in the source is more complex.
2. Please provide a clearer description of how the muscle forces were obtained. The values given in Figure 2 significantly deviate from common values found in the literature. The two references cited for the muscle model are not strongly correlated with the muscle force values obtained in this study. Furthermore, please clarify how the positions of the muscles were determined.
3. The label in Figure 2 is confusing and it is difficult to discern the exact relationships. It is recommended to revise the figure for better clarity.
4. It is recommended to include process images and results of the mechanical testing.
5. The sequence of the figures and tables is inconsistent with their order of appearance in the main text. It is suggested to insert the graphics immediately after the first mention in the corresponding paragraphs, as per the journal's guidelines.
6. Figure 4 has poor quality and insufficient clarity. It is recommended to improve the resolution.
7. It is recommended to include images of Models 1, 2, 3, and 4 to demonstrate the model construction process. Alternatively, Figure 1 could be revised to make the model construction process clearer. Additionally, it is advised to include figures of the finite element models with meshes.
8. Please provide details on the specific settings between the teeth and the mandible. Furthermore, include a figure of the contact region for the surface-to-surface contact settings.
9. It is recommended to add a comparison figure of the two workflows, such as material property maps, finite element mesh figures, etc.
10. Although a dental model has been established, the periodontal ligament has not been considered. What is the significance of the tooth modeling?
11. Lines 367-374 contain repetitive descriptions. It is recommended to remove them.
12. The description in lines 376-382 is not a result. Please revise accordingly.
13. The results section of the manuscript mostly repeats figure legends and lacks results with clear numerical values and significance.
14. Please specify the units when describing stress values.
15. Please standardize the creation of stress contour maps.
16. The manuscript lacks a clear focus, with disorganized descriptions and a lack of reliable, meaningful information.
Author Response

(The authors gave the same response as above.)

Reviewer 4 Report
Comments and Suggestions for Authors
This paper presents the design of a patient-specific temporomandibular joint (TMJ) using both academic and industrial approaches under different conditions. The study is promising, as it suggests potential improvements to industrial methods by integrating academic techniques to enhance patient outcomes. After reviewing the paper, several issues have been identified that need to be addressed:
1. Abstract: The authors should provide more details on the comparison between the academic and industrial approaches, particularly if the differences can be quantified. Phrases such as “minor variations” and “these findings” are too vague. Providing more specific information would improve the paper’s clarity and visibility.
2. Introduction: At the end of this section, the structure of the subsequent sections should be outlined to aid the reader’s understanding of the paper.
3, Section 2 (Patient Data):
a). Please clarify whether ethical approval and informed consent were obtained for the patient involved in the research.
b). Specify which software was used for segmentation and model generation in Figure 1 (A–D) and (E–F), as different software appears to be used according to the content.
c). The last paragraph on page 9 and the description of Figure 3 are somewhat repetitive.
4. Section 3 (Wear Test & Setup):
a). Please include the specific equation used to calculate the wear rate shown in Figure 4.
b). Label all devices in the static compression test setup (Figure 5A), especially the load cell.
c). The paragraph between lines 398–406 on page 12 seems misplaced, as Figure 1 has already been presented and described in Section 2.
d). Similarly, the last paragraph (lines 430–436) on page 13 repeats information already discussed in Figure 3.
5. Section 4: This section should be divided into Discussion and Conclusion. The Conclusion should focus solely on the key results and outcomes of the study.
Author Response
Dear reviewer,
we answered everything in the attached pdf file.

Round 2
Reviewer 1 Report
Comments and Suggestions for Authors
I thank the authors for responding to my comments. The Discussion reads well now with its comment about the potential links between this work and clinical management.
I am happy with it and recommend acceptance for publication.
Author Response
We sincerely thank the reviewer for the thoughtful comments and constructive suggestions, which have improved the quality and clarity of our manuscript. We are especially pleased that the revised Discussion now effectively communicates the potential clinical implications of our work. We appreciate your positive feedback and recommendation for acceptance.
Thank you once again for your time and support throughout the review process.
Kind regards,
The Authors
Reviewer 2 Report
Comments and Suggestions for Authors
- Please incorporate the author's response to my previous "Academic workflow" comment in the introduction. (We appreciate the reviewer’s observation regarding the term “academic workflow.” In our manuscript, “academic workflow” refers to the simulation process that is based exclusively on image segmentation and post‐operative CT data rather than on proprietary pre-operative CAD data.)
- Similarly, the response to the comment on patient selection should be mentioned as a limitation. (The patient included in this report was selected because she underwent a unilateral TMJ replacement that provided both comprehensive pre-operative and post-operative imaging data. Furthermore, she was one of the first patients that received a customized TMJ prosthesis and therefore, we were not able to increase the number of cases now. In future, we want to do a simulation-based study with a significant number of patients to do also some statistics. We agree with the reviewer that this is of importance and should be done if a relevant number of cases is available.)
Author Response
We answered all points of the reviewer in the attached pdf file.

Reviewer 3 Report
Comments and Suggestions for Authors
The new revision does not eliminate my previous concerns. While the original intent of the study is interesting, it needs reliable methods and clear results to be convincing. I recognize that the authors made a lot of effort to respond. However, there were fewer improvements in the revised manuscript. Many of the changes to the problem are not reflected in the revised manuscript. In addition, even for the same model, the simulation results in different software can be somewhat different.
Author Response
We regret that we were only able to partially address the reviewer's concerns. Nevertheless, we sincerely appreciate the time and effort they dedicated to reviewing our manuscript, as the reviewer’s feedback has undoubtedly improved it, despite the remaining issues.
Thank you once again for your time and support throughout the review process.
Kind regards,
The Authors
Reviewer 4 Report
Comments and Suggestions for Authors
The previous comments have been thoroughly addressed, and the quality of the paper has been significantly improved. I have no further comments.